# Reorganization of Parvalbumin Immunopositive Perisomatic Innervation of Principal Cells in Focal Cortical Dysplasia Type IIB in Human Epileptic Patients

**DOI:** 10.3390/ijms23094746

**Published:** 2022-04-25

**Authors:** Cecília Szekeres-Paraczky, Péter Szocsics, Loránd Erőss, Dániel Fabó, László Mód, Zsófia Maglóczky

**Affiliations:** 1Human Brain Research Laboratory, Institute of Experimental Medicine, ELKH, 1083 Budapest, Hungary; paraczky.cecilia@koki.hu (C.S.-P.); szocsics.peter@koki.hu (P.S.); 2Szentágothai János Doctoral School of Neuroscience, Semmelweis University, 1085 Budapest, Hungary; 3Department of Functional Neurosurgery and Center of Neuromodulation, National Institute of Mental Health Neurology and Neurosurgery, 1145 Budapest, Hungary; l.g.eross@gmail.com (L.E.); fabo.daniel@gmail.com (D.F.); 4Department of Pathology, St. Borbála Hospital, 2800 Tatabánya, Hungary; modlaszlo52@gmail.com

**Keywords:** perisomatic inhibition, dysmorphic neurons, focal epilepsy, human cortex, parvalbumin, FCD, dysplasia, interneurons, inhibitory function, neurodevelopmental disorder

## Abstract

Focal cortical dysplasia (FCD) is one of the most common causes of drug-resistant epilepsy. As several studies have revealed, the abnormal functioning of the perisomatic inhibitory system may play a role in the onset of seizures. Therefore, we wanted to investigate whether changes of perisomatic inhibitory inputs are present in FCD. Thus, the input properties of abnormal giant- and control-like principal cells were examined in FCD type IIB patients. Surgical samples were compared to controls from the same cortical regions with short postmortem intervals. For the study, six subjects were selected/each group. The perisomatic inhibitory terminals were quantified in parvalbumin and neuronal nuclei double immunostained sections using a confocal fluorescent microscope. The perisomatic input of giant neurons was extremely abundant, whereas control-like cells of the same samples had sparse inputs. A comparison of pooled data shows that the number of parvalbumin-immunopositive perisomatic terminals contacting principal cells was significantly larger in epileptic cases. The analysis showed some heterogeneity among epileptic samples. However, five out of six cases had significantly increased perisomatic input. Parameters of the control cells were homogenous. The reorganization of the perisomatic inhibitory system may increase the probability of seizure activity and might be a general mechanism of abnormal network activity.

## 1. Introduction

The most common symptom of focal cortical dysplasia (FCD) is the appearance of epileptic seizures [1,2,3]. The abnormal brain activity characteristic of seizures is caused by the synchronous firing of neuronal populations, which can be related to the inadequate functioning of the inhibitory system [1,4,5,6,7]. Epileptic activity has been associated with the structural and synaptic reorganization of the glutamatergic and GABAergic (GABA: gamma-aminobutyric acid) systems in the hippocampus (HC) [6,8,9,10,11] as well as in the neocortex [12,13,14] of both animal models and humans. About 30–40% of epileptic patients are drug resistant [15,16,17], and these cases frequently have an FCD background. In a 2017 analysis, FCD was the most common diagnosis among surgically treated epileptic children [18,19]. Furthermore, according to statistical data from 16 European epilepsy centers in 2018, a higher number of recent cases was associated with dysplasia than with hippocampal sclerosis (HS) in drug-resistant subjects [20]. Thus, investigating the pathomechanism of FCDs has become an increasingly important and relevant task.

Cortical dysplasias are caused by neurodevelopmental disturbances. Type I FCDs are characterized by abnormal radial and/or tangential cortical lamination and minor cellular abnormalities (type IA, IB, IC). Type II FCDs show severe local dyslamination and abnormal cell types, such as dysmorphic neurons (type IIA) and balloon cells (type IIB). Type III FCDs are identified by cortical lamination abnormalities adjacent to various types of other lesions. Balloon cells specifically occur only in type IIB [3,21,22,23,24].

The presence of cell types expressing immature markers in FCDII suggests that the deviation from normal development begins in the proliferation phase [2,25]. The genetic results suggest that damage of the mammalian target of rapamycin signaling pathway plays a major role in the development of this impairment [1,26,27,28,29]. Accordingly, common features of type IIB FCD are varying degrees of disruption of the cortical lamination, the presence of balloon cells, abnormal glial cells, cytomegalic interneurons, as well as hypertrophic and dysmorphic neurons [1,30,31]. These aberrant cell types are characterized by abnormally large cell bodies and abnormal processes [3,21,23,32]. They can be detected with immunostaining against mature and immature neuron markers, such as nestin, vimentin, class III β-tubulin, neuronal nuclei (NeuN), and non-phosphorylated neurofilament H (SMI32), but not with mature glial markers, e.g., glial fibrillar acidic protein (GFAP) [1,19,32,33]. Balloon cells may contain various combinations of these markers as well as GFAP [1,32].

Our present study focused on the perisomatic input of layer III and V principal cells in FCDIIB patients. Here, we refer to principal cells exhibiting aberrant dendritic arborizations and enlarged cell bodies in FCD samples as “giant” principal cells. These cells are most likely modified pyramidal cells (presumably dysmorphic neurons) [1,34,35] and most of them are immunopositive for SMI32 and NeuN [1,19,32]. Although balloon cells also possess large cell bodies and may contain SMI32 [32], their localization (mostly white matter and layer VI) is dominantly out of our sampling area [1,34,35]. Thus, it is unlikely that they are among the measured cells.

Immunohistochemical studies of human tissue with FCDIIB yielded several conflicting results [14,25,31,36,37,38,39,40,41]. This was most likely caused by different study parameters, such as the comparison of epileptic tissues with different pathologies without control, or using control samples with long postmortem delay and different localization, among others.

Epileptic samples often show mixed pathologies or originate from different brain regions than the control tissue [25,31,36]. By comparing control and FCDIIB epileptic samples from different cortical regions, Nakagawa et al. found a significant decrease in the number of parvalbumin (PV)-immunopositive cells [25]. In contrast, other researchers found no difference in the number of PV-immunopositive elements in FCDIIB samples from different cortical regions with various pathology compared to “cryptogenic” (no visible lesion) surgical samples [31]. Kuchukhidze et al. found elevated amounts of PV-immunostained cells and fibers in FCD cases in various mixed-grade samples, cryptogenic samples, temporal lobe epilepsy (TLE) samples compared to control tissues with relatively long post-mortem delay (10 h, range: 8–13) [36]. In summary, the FCD cases are characterized by the alteration of GABAergic system, but available data are inconclusive about the quantitative changes due to a large variation in sampling methods and post-mortem interval (PMI) of studied tissue samples.

It is also common practice to use the cortical tissue nearby dysplastic lesions or tumors (cryptogenic tissue) for control in research projects [14,31,41,42]. In a previous study, Rossini et al. found no significant dendritic and synaptic changes in tissues adjacent to the lesions compared to autopsy samples examining at the light microscopic level. However, relatively long PMI (i.e., autopsies performed after within 48 h after death) [14] may cause severe dendritic decomposition, for example [8,43,44,45]. Moreover, it has been reported that, in the vicinity of tumors, the blood–brain barrier and the perineural network are impaired [46,47]. This can lead to the dysfunction of GABAergic cells, reducing the neural inhibition while releasing a large amount of glutamate. These conditions probably affect the surrounding cortical areas, as well [46,47,48,49].

The above examples show the difficulty of sampling the same cortical areas with the same pathological conditions and their comparison with “valid” (neurological disease/lesion-free) controls from the same cortical areas of human patients. Accordingly, our aim was to compare cortical samples with matching pathology to appropriate control tissues with short post-mortem intervals.

Our previous investigations were performed on HC surgical samples of patients with TLE. We mainly identified preserved or sprouting perisomatic inhibitory input, which may increase the synchronous firing of principal cells and seizure probability [6,10,50,51]. Therefore, we wanted to investigate the changes of perisomatic inhibition in FCD IIB samples. Somatic coverage of PV-immunopositive terminals on individual principal cells was examined in FCD cases and compared to carefully selected, appropriate control samples with short post-mortem delay to decrease the variability of parameters.

## 2. Results

### 2.1. Dependence of Immunostaining on Fixation and Post-Mortem Delay

According to previous investigations, for electron microscopic and quantitative studies, tissue samples were chosen with a post-mortem delay of less than 4 h since their ultrastructural preservation is similar to the surgical samples [51,52,53,54]. The PV stainings in the tissue samples included in the study were similar to each other and to previously described human or monkey PV-immunostaining in the neocortex [55,56]. The effect of the age of the included subjects was also examined, and older subjects showed similar preservation and immunostaining for PV to younger ones. Therefore, we suppose that differences in PV-immunostaining between our control and epileptic samples may most likely be caused by FCD pathology.

### 2.2. Cortical Structure and Examined Cell Types in Control and FCD Samples

Well-preserved post-mortem control samples (*n* = 6) with short PMI were selected, and preservation criteria were based on our earlier studies [50,57].

The principal cells were visualized by SMI32-immunostaining (Figure 1) to examine their size and distribution in the samples. The distribution of the astrocytes was investigated by GFAP-immunostaining (Figure 2). In the control samples, the cortical layers were normal based on SMI32-immunostaining (Figure 1A), while principal cells and astrocytes showed normal distribution and morphology (Figure 1A and Figure 2A). The perisomatic input was examined based on PV-immunostaining (Figure 3) and quantified in NeuN-PV double immunostained sections using a confocal fluorescent microscope (Figure 4C,D). PV-immunopositive interneurons were present throughout the cortex. The PV-immunopositive interneurons in both the control and epileptic tissue samples with FCDIIB were heterogeneous and multipolar (Figure 3). In the control cases, the densest PV-immunopositive fiber network was located from the bottom of layer III to the top of layer V [55,56,58] (Figure 3A).

The pathological pattern of 16 epileptic patients with FCD varied from control-like tissue samples to disorganized cortical layers and abnormal cells. We have selected samples with FCDIIB dysplasia in the prefrontal (BA46, four patients), parietal (BA7, one patient), and occipital (BA18, one patient) cortices (Table 1). These six FCD subjects with the same pathological diagnosis, FCDIIB [2,3,23,59,60], underwent quantitative analyses. In the FCDIIB samples, cortical lamination showed heterogeneity. Usually, it was slightly abnormal, with local dyslamination. For the quantitative measurements, regions with preserved lamination were selected and compared to control tissue (Figure 1B and Figure 3B).

An important feature of the FCDIIB pathology is the presence of dysmorphic and hypertrophic neurons with abnormal morphology and enlarged cell bodies in the cortex [1,3,21,24]. Cells with abnormal size and morphology were found in our FCDIIB samples (Figure 1B,E, Figure 3E,F and Figure 4D), they are referred to as “giant” principal cells. The giant neurons were present mostly in layers III and V (Figure 1B,F, Figure 3E and Figure 4D).

The pathology of FCDIIB can be very diverse, but its constant feature is the presence of balloon cells in layer VI and the white matter [3,21,32,34]. The white matter in our FCDIIB cortices has shown presumable balloon cells with rounded bodies and few thin processes (Figure 1E and Figure 2E).

The FCDIIB samples differed from each other: HE117, HE177, HE180, and HE239 were somewhat similar to the controls in cell size and morphology, with a low number of giant neurons and presumable balloon cells; whereas HE199 and HE220 displayed numerous pathological giant cells with abnormal processes and a large number of presumable balloon cells in the white matter and layer VI (Figure 1E,F, Figure 3E,F and Figure 4D).

GFAP-labeled astrocytes in the control cortices showed a loose distribution (except in the first layer) and they were more frequent around blood vessels (Figure 2A,C). The number of GFAP-immunopositive elements was increased in the FCDIIB cortices (Figure 2B,D). Several glial cells were found with abnormally large cell bodies, and their processes were thicker and shorter (Figure 2D), showing the signs of reactive astrocytosis [21,61]. Presumable balloon cells were found in the FCDIIB samples with GFAP- immunostaining as well (Figure 2B,E). According to recent papers, balloon cells express the markers of both mature and undifferentiated cells. Furthermore, balloon cells may also express GFAP [32].

Some PV-positive interneurons in the FCDIIB cases had a much larger cell body and/or abnormal localization (Figure 3B,D). These are presumably cytomegalic interneurons [1,30,31]. More intense PV-immunostaining was observed in the FCD cases (Figure 3B). The negative profiles of giant cells were prominently highlighted by the PV-immunopositive “baskets” of presumable inhibitory terminals surrounding them (Figure 3E,F).

Considering these properties, we have investigated the NeuN-positive principal cells of layer III and layer V and the PV-immunopositive perisomatic terminals surrounding them.

### 2.3. Quantitative Analyses of PV-Immunopositive Perisomatic Terminals in the Cortices of Control and FCDIIB Cases

Perisomatic inhibition refers to synaptic input on the cell bodies and proximal dendrites originating from basket cells, or on the axon initial segments (AIS) by the axo-axonic cells [62]. In the present study, PV-immunopositive terminals contacting the cell body were counted (Figure 4C,D). In the course of quantitative analyses, the diameters of sampled cells were measured, and the number of close contacting terminals was counted. The number of PV-positive terminals was divided by soma perimeter length and the result was normalized to unit-perimeter.

The perisomatic PV-immunopositive terminals were also examined by electron microscopy. They have established symmetrical synapses on the cell bodies (Figure 5).

Table 2 shows the number of PV-immunopositive perisomatic terminals contacting the measured cell bodies per number of measured cells.

We found that the long diameters in more than 90% of the NeuN-labeled cell bodies are less than 30 µm in control samples (measured cells of layer III < 30 µm = 96.13%; measured cells of layer V < 30 µm = 98.79%). Further statistical analysis was carried out after the separation of cells according to soma length to investigate whether large cells (>30 µm long soma diameter), i.e., giant neurons, show different input properties in dysplastic cases (see below).

In the two cases with numerous abnormal neurons (HE199, HE220), the giant cells are contacted by a larger number of PV-immunopositive perisomatic terminals per unit-perimeter, compared to controls (Figure 6, Table 3). On the contrary, in these two FCD cases, we found fewer labeled terminals around the normal-sized cells (<30 µm long soma diameter) than around the giant cells in both layers, and in the controls in layer III. In layer V, the normal-sized cells were contacted by a slightly larger number of perisomatic terminals compared to controls (Figure 6, Table 3). The other four patients (HE117, HE177, HE180, and HE239) were similar to the controls regarding the cell size (Figure 6A,B). Therefore, their somatic input properties were analyzed together. We have found a significantly increased number of PV-immunopositive terminals in layers III and V in three (HE117, HE177, and HE239) out of these four cases in comparison with region-matched controls (SKO18, SKO19, SKO13, SKO20) (Figure 6, Table 4).

Data from FCDIIB and control samples were pooled per layer and the number of PV-immunopositive perisomatic terminals per unit-perimeter was compared using the Mann–Whitney *U* test. A comparison of pooled data showed that the number of perisomatic PV-immunopositive terminals per unit-perimeter is significantly higher in FCD cases: Layer III: Mann–Whitney *U* = 35,088, N1 = 336, N2 = 327, *p* = 6.89 × 10^−16^; Layer V: Mann–Whitney *U* = 41,020, N1 = 332, N2 = 331, *p* = 7.06 × 10^−9^ (Figure 4A,B).

## 3. Discussion

It is known that PV-immunopositive interneurons play an important role in the generation of gamma oscillation (30–100 Hz), which synchronizes cortical activity, and appear mainly in association with cognitive processes in the EEG pattern [62,63,64,65]. Excessive synchrony can lead to epileptic seizures and the increase in interictal gamma activity often predicts a subsequent seizure [5,66]. The role of inhibitory function in pathological synchrony has been highlighted by several experimental results [5,53,66,67,68,69].

In the present study, our aim was to compare epileptic samples with the same pathological condition to each other and to control samples of the same cortical Brodmann area. The control samples had short PMI (2–4 h), while the epileptic samples underwent pathological classification and proved to be FCDIIB cases. Despite the heterogeneous morphological features, the perisomatic inhibitory input of principal cells in FCDIIB samples is significantly increased compared to the control samples (Figure 4A,B).

Earlier studies [14,25,31,36,37,38,39,40,41] showed mixed results regarding the changes in the amount of the perisomatic input of FCD cortices. Our samples were similar to each other in showing a general increase in this property (Figure 6C,D). However, individual cells of the epileptic subjects themselves showed heterogeneity in the PV-immunostained somatic input (Figure 6). This may be the result of several factors, such as the sampling process, e.g., site of the sampling in the dysplastic area, and variance in the history of epilepsy among patients. However, perisomatic input abnormalities were found in all samples, disregarding the amount of the abnormal cell types. Our results also suggest that “control-like” tissue samples from the vicinity of epileptic lesions may differ significantly from control samples and might not be valid controls.

The extent of the functional morphological changes in FCD might also be related to the duration of epilepsy [70,71]. Some high-affinity neurotrophin receptors, e.g., tyrosine kinase receptors A, B, C, show significantly elevated expression on dysmorphic and balloon cells in FCD-affected tissue specimens [42]. Neurotrophin levels are often elevated by seizures, and this can cause cell hypertrophy, among others [42]. One of the neurotrophin receptors, p75, is involved in the regulation of connections of PV-containing inhibitory cells as a negative regulator, so the elevated p75 ligand level due to seizures may result in the downregulation of the inhibitory function of PV-containing cells [72].

In four subjects out of six, mostly control-sized principal cells were found (Figure 6A,B). However, in three out of four cases, they showed more numerous PV-immunopositive somatic terminals compared to matched controls (Figure 6, Table 4). Besides the above-mentioned causes of result variability, prolonged epileptic disorder and medication might also affect the reorganization of inhibitory connections of the FCD-affected tissue samples. Although some results suggest that the duration of epilepsy alone would not cause such differences [14], the contribution of this factor cannot be ruled out either [70,71].

In two cases (HE199, HE220) with numerous giant neurons and presumable balloon cells, the giant cells have more inhibitory inputs compared to controls (Figure 6). On the contrary, around the normal-sized cells, we found fewer labeled terminals than around the giant cells in both layers, and in the controls in layer III (Figure 6C,D, Table 3). These maldeveloped pyramidal cells or dysmorphic neurons may be involved in the generation of epileptogenic activity, as has been described in several studies [1,34,35]. Furthermore, the largest differences among the input characteristics were found in layer III, suggesting that cortico-cortical connections of principal cells may be involved in increased electric activity, too [73].

The abnormal function of giant neurons may also be related to their abnormal perisomatic innervation by PV-immunolabeled terminals. Namely, increased inhibition may enhance synchronization, which further increases the chance of seizure formation [5,53,66,67,68,69]. The affected area may also develop into a stimulatory focus if the effect of GABA becomes excitatory due to pathological conditions [4,74]. A similar pattern of perisomatic input reorganization was described in the HC of TLE patients [6,10,50].

### Limitations

It is not known how many of the studied perisomatic terminals form synapses with the cell body. However, it has been shown that the terminals around the cell body most likely form functional synapses [50]. Accordingly, there is a good chance that the examined terminals here also formed synapses. Using the electron microscope, we identified symmetric synapses between the PV terminals and cell bodies (Figure 5). Therefore, we may assume that at least a considerable amount of the close contacts refers to synaptic input.

In this study, only the terminals on the cell bodies were quantified. The terminals around the proximal dendrites and on the AIS are also part of the perisomatic inhibition [62,75]. Based on our light microscopic observations, an increase in the PV-immunopositive input can be seen on the dendrites, too. Further studies are needed to determine the quantitative changes.

In addition to PV-containing cells, cholecystokinin-expressing inhibitory cells (cannabinoid receptor type 1—immunopositive) also play a role in perisomatic inhibition [10]. Further investigations are needed to assess their alterations in their inhibitory connectivity.

## 4. Materials and Methods

### 4.1. Obtaining the Human Tissue

Control subjects (*n* = 9) died from causes unrelated to any brain disease and the clinical data or the autopsy did not show any signs of neurological disorders. The study was approved by the ethics committee at the Regional and Institutional Committee of Science and Research Ethics of Scientific Council of Health (ETT TUKEB 15032/2019/EKU) and performed in accordance with the Declaration of Helsinki. The control brains were removed 2–5 h after death, the internal carotid and vertebral arteries were cannulated, and the brains were perfused first with physiological saline (1.5 L in 30 min) containing 5 mL of heparin, followed by a Zamboni fixative solution containing 4% paraformaldehyde, 0.05% glutaraldehyde, and 15% picric acid in phosphate buffer (PB, pH 7.4) (4–5 L in 1.5–2 h). After perfusion, 0.5–1 cm thick blocks were cut from the cortical regions of the brain. Regions were identified according to Brodmann division [76], i.e., Brodmann areas 7, 18, 38, and 46, and post-fixed in the Zamboni solution without glutaraldehyde overnight [57]. For quantitative analyses, six cortical samples of five control subjects were selected (Table 1).

Patients (*n* = 16) with drug resistant epilepsy accompanied with focal cortical dysplasia (FCDI-II-III) in conformity with the International League Against Epilepsy classification [60] underwent surgery in the National Institute of Mental Health Neurology and Neurosurgery in Budapest, Hungary within the framework of the Hungarian Epilepsy Surgery Program. Written informed consent for the study was obtained from every patient before surgery. The seizure focus was identified by multimodal studies, including video-EEG monitoring, magnetic resonance imaging, single photon emission computer tomography, and/or positron emission tomography. The identified epileptic tissue samples, namely the frontal (BA46 equivalent), parietal (BA7 equivalent), occipital (BA18 equivalent), and temporal (BA38 equivalent) cortical areas, were surgically removed. After surgical removal, the epileptic tissue was immediately cut into 0.5–1 cm thick blocks and immersed into a Zamboni fixative containing 4% paraformaldehyde, 0.05% glutaraldehyde, and 15% picric acid in 0.1 M PB (the same fixative solution used for the control brain perfusion). Fixative was changed hourly to a fresh solution during constant agitation for 6 h, and the blocks were then post-fixed overnight, in the same fixative without glutaraldehyde [57]. For quantitative analyses, six FCDIIB patients were selected (Table 1).

Then, the samples were washed in 0.1 M PB and kept in a 30% sucrose solution for 2 days, before being frozen over liquid nitrogen and stored at −80 °C. Subsequently, 60 µm thick parallel sections were prepared from the blocks with a Leica VTS-1000 Vibratome for immunohistochemistry. The 60 µm thick sections were cut from the blocks, and subsequently washed in PB 4 times, then immersed in 30% sucrose for 1–2 days and freeze-thawed three times over liquid nitrogen.

### 4.2. Immunohistochemistry

Sections were processed for immunostaining as follows. After thoroughly washed in 0.1 M PB, endogenous peroxidase activity was blocked by 1% H_2_O_2_ in TRIS buffered saline (TBS, pH 7.4) for 10 min. TBS was used for all the washes (3 × 10 min between each antiserum) and for dilution of the antisera. Non-specific immunostaining was blocked by 4% bovine serum albumin. We used antibodies against NeuN (1:2000 Chemicon) and SMI32 (1:4000, Biolegend—both antisera raised in mice) to detect the principal cells. To stain the perisomatic inhibitory elements, PV (1:5000, Swant, raised in mouse and rabbit antibody), and for labeling the astrocytes, GFAP (1:2000, Chemicon, raised in mouse) were used. All primary and secondary antibodies were tested by the producers for specificity. Sections were incubated for 1 day at room temperature or 2 days at 4 °C. For the light and electron microscopic visualization of the immunopositive elements, biotinylated anti-rabbit/mouse IgG (1:250, Vector) was applied as secondary serum followed by avidin-biotinylated horseradish peroxidase complex (1:250, Vector). Sections were incubated in 3,3′-Diaminobenzidine tetrahydrochloride (DAB, Sigma, St. Louis, MO, USA) as a chromogen dissolved (in TRIS buffer) and the immunoperoxidase reaction was developed by 0.01% H2O2. Sections were then treated with 0.5% osmium tetroxide in PB for 10 min. Dehydration comprised a 50% to absolute ethanol series, with an additional step of uranyl acetate at the 70% ethanol stage for 30 min and mounted in Durcupan (ACM, Fluka, St. Louis, MO, USA). Samples were analyzed using conventional optical and electron microscopy.

For the confocal microscopic investigations, we used the same methodology in the preparation of sections as listed above until the incubation of primary antibodies. Incubation with both primary antibodies took place simultaneously (anti-NeuN raised in mouse 1:2000, anti-PV raised in rabbit 1:5000). To aid the penetration of primary antibodies, 0.1% triton was added to the blocking solution. After the incubation, secondary antibodies with fluorophores were applied (Alexa488 donkey anti-mouse 1:500, Alexa594 donkey anti-rabbit 1:500, Molecular Probes, Eugene, OR, USA) for 3 h. For the reduction of autofluorescence, we incubated the samples with 3 mmol/L copper sulfate and 50 mmol/L ammonium acetate (pH 5) solution for 40 min [77]. Then, we mounted them in Vectashield antifade medium (Vector).

### 4.3. Electron Microscopy

PV-DAB immunolabeled sections were prepared for qualitative electron microscopic analysis. Areas of interest in layer III were re-embedded and 60 nm ultrathin slices were prepared with an ultramicrotome (Leica EM UC6) and collected onto Formvar-coated single-slot copper grids, then counterstained with lead citrate (Ultrostain, Leica, Wetzlar, Germany). The samples were investigated using a transmission electron microscope (Hitachi H-7100) and the perisomatic synapses labeled by the chromogens were identified.

### 4.4. Confocal Microscopy, Quantitative Analysis

For quantitative analyses, equivalent cortical samples of six control subjects and six FCDIIB patients were selected (Table 1). The FCDIIB cortical areas were four frontal samples equivalent to BA46, one pariettoquivalent to BA7, and one occipital sample equivalent to BA18 cortices. NeuN-PV double fluorescent immunostained sections were prepared and examined in the confocal microscope in order to count the perisomatic PV-positive terminals around layer III and V NeuN-immunostained principal cells. Confocal microphotographs were taken using Nikon C2 Confocal Microscope with 4× air (Plan Fluor NA = 0.13, WD = 17.2 mm, FOV = 3215.36 µm) and 60× oil (Plan Apo VC NA = 1.45, WD = 0.13 mm, FOV = 215.04 µm) objectives. As mentioned above, for correct statistical analysis, quantitative measurements were performed in regions of the samples containing distinguishable layers (Figure 1). The regions of interest (ROIs) contained the entire width of the cortical layers, and the horizontal extents were around 250–350 µm. The longitudinal and horizontal diameters of principal cell bodies in their largest perikaryal extent of the corresponding layer were measured in µm. Soma size and perimeter were calculated from these two parameters. The ellipse formulas for area and perimeter were used:

Perimeter: $P\approx \pi [3\left(a+b\right)-\sqrt{\left(3a+b\right)\left(a+3b\right)}$; Area: A=a·b·π, where “*P*” refers to the perimeter, “*a*” refers the half of the long diameter, and “*b*” refers to the half of the short diameter.

PV-immunopositive perisomatic terminals contacting the largest perikaryal extent of the cell bodies of NeuN-labeled cells were counted in the ROI areas. Contact was determined if close contact was observed and no hiatus was visible between the terminal and the cell body. Two independent observers were involved in the counting process. PV-NeuN double-positive cells were excluded from the quantitative analyses.

The number of PV-immunopositive inputs was specified in relation to the unit length of the cell perimeters. The data of the measured cells in layer III and layer V were kept separately.

For the measurements, we used the NIS-Elements 5.01.00 program’s analytical features. For the calculations and the diagrams, we used Microsoft Excel 2016 and Statistica 13.4. For the comparison of control and FCD cortex, non-parametric Mann-Whitney *U* tests were used with a significance level of *p* < 0.05.

## 5. Conclusions

Earlier observations showed that alterations of the inhibitory system contribute to epileptic dysfunctions in FCD. According to our results, the amount of PV-immunopositive terminals is preserved or increased on cell bodies of patients with FCDIIB, except for the control-sized neurons of the cortical samples with giant cells (Figure 4C,D and Figure 6C,D, Table 3). The PV-immunopositive perisomatic terminals examined by electron microscopy formed symmetrical synapses (Figure 5). Therefore, these results may suggest the sprouting of the inhibitory axons in FCDIIB cases. The amount of PV-containing inhibitory elements may change as a result of a pre-existing developmental disorder, or as an adaptive mechanism balancing the seizures. The change of the perisomatic inhibitory system could further increase the probability of seizure activity in both cases. Therefore, the alteration of the perisomatic inhibition of principal cells may be a general mechanism of abnormal network activity in epilepsy [4,6]. In summary, our data highlight the likelihood of increased inhibitory input being a key player in seizure generation.

Furthermore, large inhomogeneity in the individual input characteristics of principal cells was found regarding the giant cells and their close vicinity, suggesting that the imbalance of the inhibitory inputs in microcircuits can also participate in seizure activity.

## Figures and Tables

**Figure 1 ijms-23-04746-f001:**
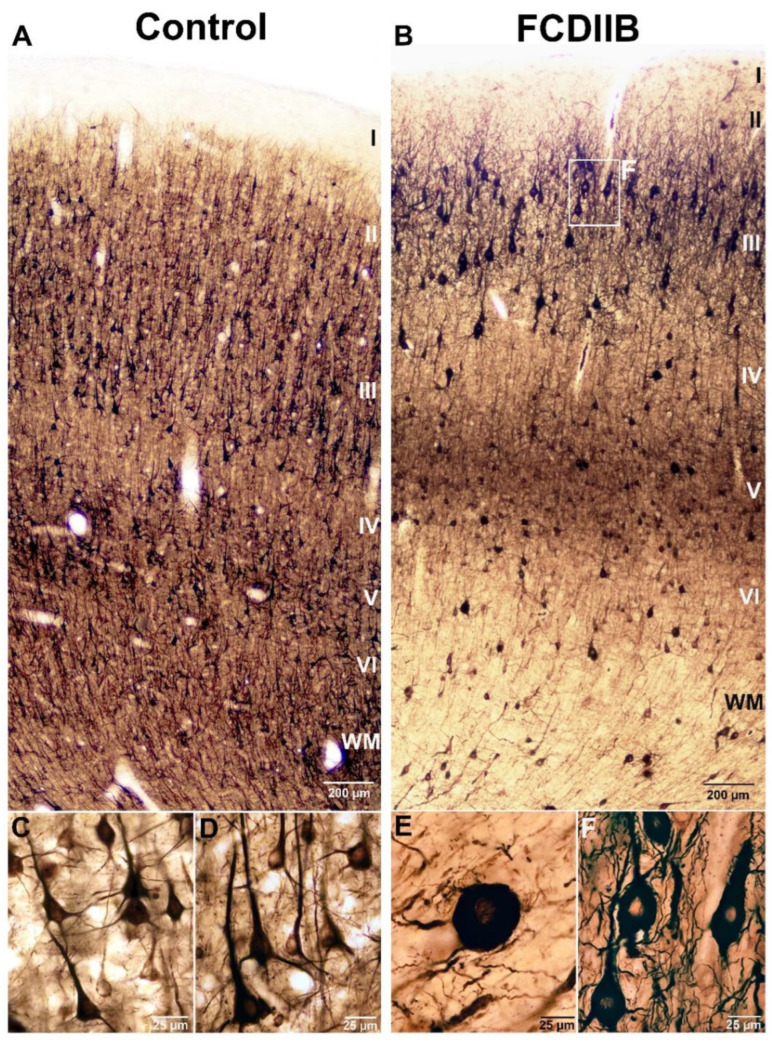
Non-phosphorylated neurofilament H (SMI32)-immunopositive elements in a control sample (**A**,**C**,**D**) and in an epileptic patient with FCDIIB in frontal cortex (**B**,**E**,**F**). SMI32-immunostaining labels principal cells with high specificity. ((**A**,**D)**: the cortical layers are marked by Roman numerals). Control principal cells from layers III and V are shown in (**C**,**D**) panels, respectively. There are numerous dysmorphic neurons in the FCD cortex ((**B**,**F**): layer III). The giant neurons have abnormally thick dendrites and thin, irregular, short processes (**F**). Presumable balloon cells are visible in the white matter and layer VI (**B**,**E**). Balloon cells have only a few, atrophic processes and large, balloon-like rounded cell bodies (**E**). Scales: (**A**,**B**): 200 µm; (**C**–**F**): 25 µm.

**Figure 2 ijms-23-04746-f002:**
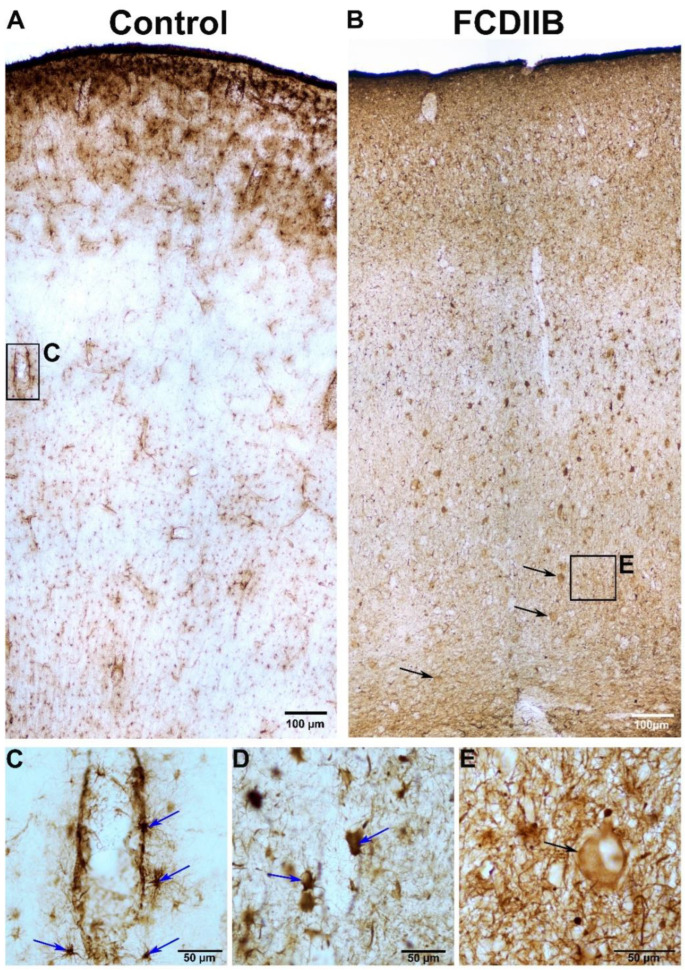
Glial fibrillary acidic protein (GFAP)-immunopositive elements in a control sample (**A**,**C**) and in an epileptic patient with FCDIIB in the frontal cortex (**B**,**D**,**E**). In the control sample, GFAP-labeled glial cells show loose distribution except in the first layer (**A**) and they are more frequent around blood vessels (**A**,**C**). Blue arrows show the GFAP-immunopositive glial cells (astrocytes). The amount of GFAP-immunopositive elements are increased in the dysplastic cortex (**B**,**D**). Both the number of cells and-their processes are also increased (gliosis) (**B**,**D**) compared to the control sample (**A**,**C**). The astrocytes are abnormally large, their processes are thicker and shorter (**D**). There are presumable balloon cells in the white matter and layer VI (**E**). Black arrows show the GFAP-immunopositive presumable balloon cells (**B**,**E**). Scales: (**A**,**B**): 100 µm (**C**–**E**): 50 µm, focused images.

**Figure 3 ijms-23-04746-f003:**
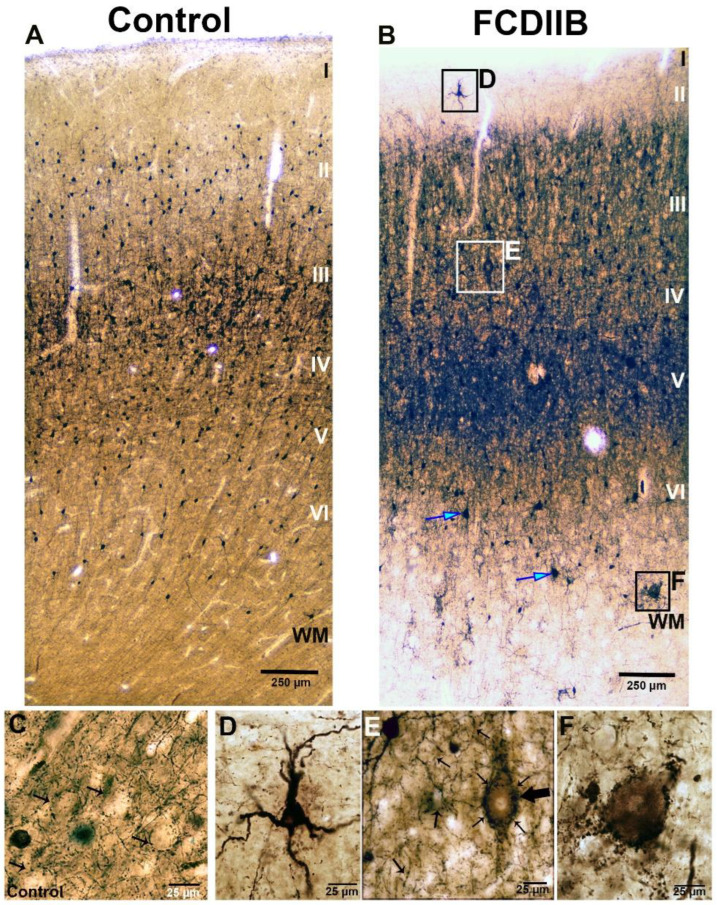
Parvalbumin (PV)-immunopositive elements in a control sample (**A**,**C**) and in an epileptic patient with FCDIIB in the frontal cortex (**B**,**D**–**F**). PV is present in a subgroup of perisomatic inhibitory cells. The maximum fiber density is found from layer III to layer V, both in the control and in the epileptic subjects ((**A**,**B**): the layers marked by roman numbers). Numerous interneurons are present in both subjects, they are heterogeneous and multipolar, and smaller in number in the epileptic cortex (**B**). Presumable dysmorphic interneurons are also present in the dysplastic cortex ((**B**): blue arrows, (**D**)). The fiber density is higher in the dysplastic cortex (**B**), the dysmorphic neurons ((**E**), large arrow) are surrounded by extremely dense PV-immunopositive terminals ((**E**), small arrows). The small arrows show the negative cells—presumably principal cells—surrounded by PV-immunopositive baskets in control and epileptic samples (**C**,**E**). Panel F shows the hypertrophic PV-immunopositive basket formation in the white matter around a presumable hypertrophic cell. Scales: (**A**,**B**): 250 µm; (**C**–**F**): 25: µm.

**Figure 4 ijms-23-04746-f004:**
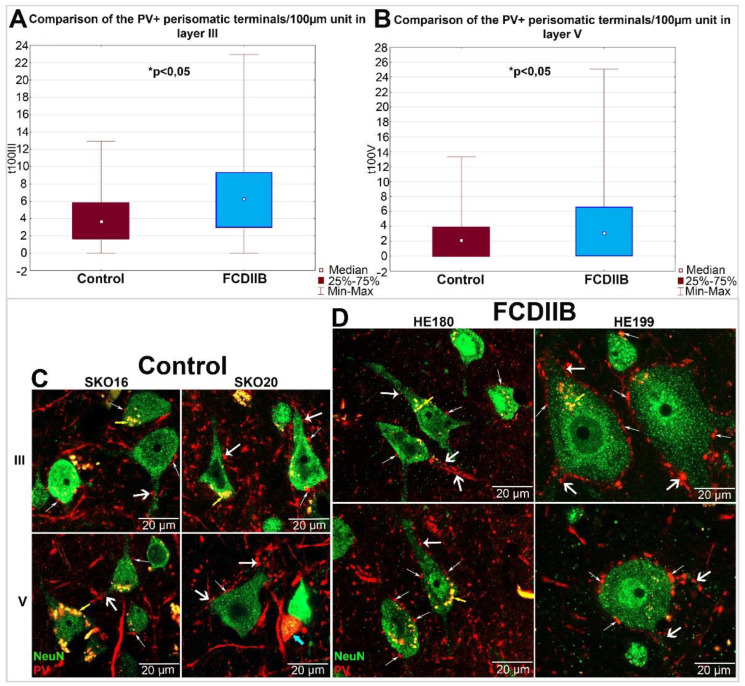
(**A**,**B**): Boxplots show a statistical comparison of the number of PV-immunopositive perisomatic terminals per 100 µm length of soma perimeter in control (claret box) and FCDIIB (blue box) subjects in layer III (**A**) and layer V (**B**) by Mann-Whitney *U* test. Both layers show significant difference in the two measured populations. Layer III: Mann-Whitney *U* = 35,088, N1 = 336, N2 = 327, *p* = 6.89 × 10^−16^; Layer V: Mann-Whitney *U* = 41,020, N1 = 332, N2 = 331, *p* = 7.06 × 10^−9^ (**A**,**B**). The increase is higher in layer III than in layer V. N1: the number of counted cells in “HE” samples; N2: the number of counted cells in “SKO” samples C, D: NeuN-PV double-immunostained cells in the prefrontal cortex of control (**C**) and FCDIIB (**D**) subjects in layers III and V with confocal fluorescence imaging. NeuN-immunopositive cells are green, PV-immunostained inhibitory cells (blue arrow), fibers and terminals are red. The perisomatic terminals are labeled with small, the terminals of the apical dendrites with narrow white arrows. The thick white arrows show the terminals at basal dendrites. Smaller and larger principal cells are mixed in layers III and V. However, abnormal giant principal cells are present in the HE199 subject in a considerable number. Yellow arrows show the autofluorescent lipofuscin drops. Scales: (**C**,**D**): 20 µm.

**Figure 5 ijms-23-04746-f005:**
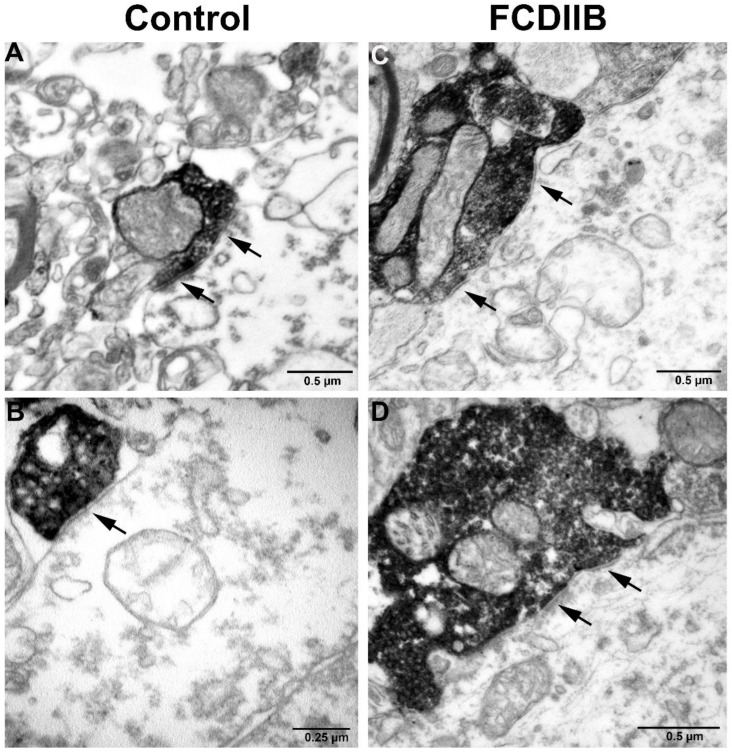
Parvalbumin (PV) immunostained sections of the somata of principal cells in prefrontal cortex of control (**A**,**B**) and FCDIIB (with abnormal cell types, **C**,**D**) subjects in layer III by electron microscopy. The PV-immunopositive perisomatic terminals are establishing symmetrical synapses with the principal neurons (arrows). Scales: (**A**,**C**,**D**): 0.5 µm, (**B**): 0.25 µm.

**Figure 6 ijms-23-04746-f006:**
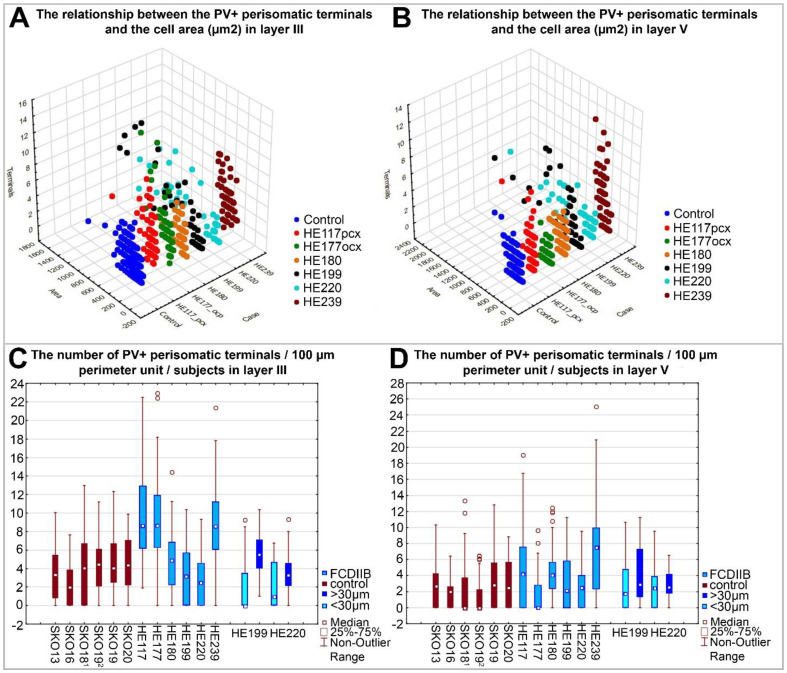
The soma area of individual NeuN-immunopositive cells and the number of PV-immunopositive perisomatic terminals contacting them are illustrated with a 3D scatterplot diagram in layer III (**A**) and layer V (**B**). Control cases are pooled due to minor variance and easier perspicuity. In the HE199 and HE220 FCDIIB samples the large neurons have larger input (black and light blue points). In the HE117, HE177 and HE239 samples the cell’s size is mostly control-like, with considerably increased number of PV-immunopositive perisomatic inputs (red, green and claret points) in layer III (**A**). Similar, but less considerable changes are visible in layer V, except for case HE239, where the input is much higher than in the control. HE180 shows control-like properties in soma size and input characteristics. The boxplots show the number of PV-immunopositive terminals calculated per unit-perimeter length (100 μm) of NeuN-immunopositive cell bodies per subject, in layer III (**C**) and layer V (**D**). Claret columns: control, blue columns: FCDIIB patients. In HE199 and HE220 FCDIIB samples the large neurons have larger input, compared to controls and the other subject with FCD. On the contrary, around the normal-sized cells (<30 μm) there are fewer labeled terminals than in the controls and around the giant cells in layer III (royal blue columns: long diameter of cells is larger than 30 µm, sky-blue columns: long diameter of cells is less than 30 µm). The neurons of the cases HE117 and HE177 have increased PV-immunopositive input in layer III (**C**). Around the cells of HE180 and HE239 there is a higher number of PV-immunopositive terminals than in control samples in both measured layers. Circles: outliers 1: pcx, 2: ocx.

**Table 1 ijms-23-04746-t001:** Data of the control and FCD patients involved in the quantitative analyses.

Control Case	Age (Years)	Gender	Post-Mortem Interval (Hours and Minutes)	Sampling Area	
SKO13	60	female	3:25	R BA46	
SKO16	72	male	2:22	L BA46	
SKO18	85	male	2:52	R BA7	
SKO19	61	female	2:53	R BA18, L BA46	
SKO20	27	male	3:45	L BA46	
Mean of control subjects	61	-	3:05	-	
FCD Case	Age (Years)-At the Time of Surgery	Duration of Epilepsy (Years)	Gender	Pathological Classification	Removed Brain Area
HE117	26	16	male	FCDIIB	R pcx/BA7
HE177	34.5	33	female	FCDIIB	R ocx/BA18
HE180	17	15	female	FCDIIB	R fcx/BA46
HE199	35	33.5	female	FCDIIB	R fcx/BA46
HE220	48	39	male	FCDIIB	L fcx/BA46
HE239	42	39.5	male	FCDIIB	L fcx/BA46
Mean of epileptic subjects	33.75	29.33	-	-	-

The table shows the data of the control (SKO) and FCD (HE) subjects. Areas sampled during autopsy from control and the surgically removed samples of epileptic FCD patients are also indicated according to Brodmann. L: Left; R: Right; fcx: frontal cortex; tcx: temporal cortex; pcx: parietal cortex; ocx: occipital cortex; BA: Brodmann Area.

**Table 2 ijms-23-04746-t002:** Number of cells and terminals counted in the quantitative analyses of the perisomatic input.

	Control Case	Control (Number of Perisomatic Terminals/Number of Cells)	FCD Case	FCD (Number of Perisomatic Terminals/Number of Cells)
Layer III	SKO13R BA46	104/60	HE117 pcx/BA7	255/58
	SKO16L BA46	66/58	HE177 ocx/BA18	265/61
	SKO18R BA7	105/51	HE180 fcx/BA46	121/50
	SKO19R BA18	90/54	HE199 fcx/BA46	132/48
	SKO19L BA46	118/60	HE220 fcx/BA46	98/49
	SKO20L BA46	108/53	HE239 fcx/BA46	251/61
Layer III mean number of terminals/cells/subject	-	98.5/56	-	187/54.5
Layer III mean number of terminals/1 cell	-	1.76		3.43
Layer III total number of terminals/cells/all subjects	-	591/336	-	1122/327
Layer V	SKO13R BA46	81/63	HE117 pcx/BA7	138/61
	SKO16L BA46	36/52	HE177 ocx/BA18	39/52
	SKO18R BA7	41/49	HE180 fcx/BA46	113/61
	SKO19R BA18	61/57	HE199 fcx/BA46	110/48
	SKO19L BA46	36/60	HE220 fcx/BA46	77/48
	SKO20L BA46	65/51	HE239 fcx/BA46	225/61
Layer V mean number of terminals/cells/subject	-	53.33/55.33	-	117/55.17
Layer V mean number of terminals/1 cell	-	0.96	-	2.12
Layer V total number of terminals/cells/all subjects	-	320/332	-	702/331
Total number of terminals and cells in layers III and V	-	911/668	-	1824/658

L: Left, R: Right, fcx: frontal cortex; tcx: temporal cortex; pcx: parietal cortex, ocx: occipital cortex, BA: Brodmann Area.

**Table 3 ijms-23-04746-t003:** The mean number of PV-immunopositive terminals/100 µm of soma perimeter in the two cases with giant cells compared to matched controls.

Samples	The Mean Number of PV-Immunopositive Terminals/100 µm of Soma Perimeter	
	Layer III	Layer V
HE199 and HE220 fcx/BA46 normal-sized cells ^1^	2.01 ± 2.50	2.67 ± 3.03
HE199 and HE220 fcx/BA46 giant cells ^2^	4.68 ± 2.68	3.53 ± 2.96
SKO13R, SKO16L, SKO19L, SKO20L fcx/BA46 ^3^	3.63 ± 2.94	2.19 ± 2.48

In the two cases with giant neurons (HE199, HE220), the giant cells (>30 µm long soma diameter) are contacted by a larger amount of inhibitory PV-immunopositive terminals per unit-perimeter compared to controls. On the contrary, around the normal-sized cells (<30 µm length of diameter) we found fewer labeled terminals than around the giant cells in both layers, and in the controls in layer III. In layer V the normal-sized cells are contacted by a slightly larger number of input terminals compared to controls. ^1^: <30 µm length of diameter; ^2^: >30 µm length of diameter; ^3^: matched controls; fcx: frontal cortex; BA: Brodmann Area.

**Table 4 ijms-23-04746-t004:** Comparisons of the number of PV-immunopositive terminals/100 µm of soma perimeter with Mann-Whitney *U* test in cases with mostly normal-sized cells.

Samples	Comparisons of the Number of PV-Immunopositive Terminals/100 µm of the Perimeter with Mann-Whitney *U* Test ^1^	
	Layer III	Layer V
HE117/SKO18 pcx/BA7	*U* = 595.000, N1 = 58, N2 = 51, *p* = 2.22 × 10^−8^	*U* = 922.000, N1 = 61, N2 = 49, *p* = 4.92 × 10^−4^
HE177/SKO19 ocx/BA18	*U* = 550.500, N1 = 61, N2 = 54, *p* = 8.23 × 10^−11^	*U* = 1097.000, N1 = 52, N2 = 57, *p* = 1.92 × 10^−2^
HE239/SKO13 fcx/BA46	*U* = 577.500, N1 = 61, N2 = 60, *p* = 5.39 × 10^−12^	*U* = 903.500, N1 = 61, N2 = 63, *p* = 1.53 × 10^−7^
HE180/SKO20 fcx/BA46	*U* = 1268,0000, N1 = 50, N2 = 53, *p* = 7.10 × 10^−1^	*U* = 1257,000, N1 = 61, N2 = 51, *p* = 8.17 × 10^−2^

Comparisons of the number of PV-immunopositive terminals/100 µm of soma perimeter with Mann-Whitney *U* test. In comparison with matched controls, we have found significantly increased PV-positive inhibitory input in layers III and V in 3 out of 4 cases (HE117, HE177, and HE239) and there was no difference in HE180. ^1^: the significance level is *p* < 0.05; N1: the number of counted cells in “HE” samples; N2: the number of counted cells in “SKO” samples; pcx: parietal cortex; ocx: occipital cortex; fcx: frontal cortex; BA: Brodmann Area.

## Data Availability

The data supporting the findings of this study are available on request from the corresponding author.

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
