# Peer review of "Reorganization of Parvalbumin Immunopositive Perisomatic Innervation of Principal Cells in Focal Cortical Dysplasia Type IIB in Human Epileptic Patients"

_ijms, 2022, doi:10.3390/ijms23094746_

Round 1

Reviewer 1 Report

Focal cortical dysplasia (FCD) is characterized by resistance to pharmacotherapy of epileptic seizures. Abnormal functioning of the perisomatic inhibitory system may play an important role in the occurrence of seizures. In the present work,  Szekeres-Paraczky and co-authors investigated whether there are changes in inhibitory inputs to the soma of neurons in brain tissue samples of operated patients, suffering from FCD, compared to similar sections of the cerebral cortex of dead people who did not have signs of FCD. The authors showed that in FCD patients, perisomatic inputs are significantly more abundant than in neurons of control tissuse. The number of parvalbumin-immunopositive perisomatic inhibitory terminals in contact with the principal cells was significantly greater in epileptic cases. These data suggest that the reorganization of the perisomatic inhibitory system may be the cause of abnormal network activity and increased seizure activity. 

Notes concerning to inaccuracies in the figures: 

Figure 1. Panel (A) does not contain rectangulars outlining the areas corresponding to panel C, D. Panel (B) does not contain rectangular outlining the area shown in panel E. 

Figure 2. Panel (A) does not contain rectangular outlining the area corresponding to panel C. Panel (B) does not contain rectangulars outlining the areas shown in panels  D and E.

Figure 3. Panel (A) does not contain rectangular outlining an area corresponding to panel C. Panel (B) contains blue arrows which were not mentioned in the figure legend.

Reviewer 2 Report

The Authors have described the studies carried out in great detail and illustrated them with appropriate figures and tables.

However, I have a few doubts about the choice of patients whose tissues were collected.

1) Why were these particular layers of cortex examined? Since e.g. balloon cells also characteristic for this disorder, are located in other parts of the cortex

(2) Was the frequency and type of seizures in potential patients taken into account? In particular that the initial number of patients was 16 (line 357)

(3) Has any attempt been made to find a relationship between cortical changes and seizure frequency? And also the age of the patients and the duration of epilepsy?
